# A Deep Inverse-Mapping Model for a Flapping Robotic Wing

**Hadar Sharvit**[1,2,3], **Raz Karl**[1,2,3] **& Tsevi Beatus**[1,2,3] *

[1] School of Computer Science and Engineering
[2] The Institute of Life Sciences
[3] Center for Bioengineering
The Hebrew University of Jerusalem, Israel 9190401
{hadar.sharvit1, raz.karl ,tsevi.beatus}@mail.huji.ac.il

## Abstract

In systems control, the dynamics of a system are governed by modulating its inputs to achieve a desired outcome. For example, to control the thrust of a quad-copter propeller the controller modulates its rotation rate, relying on a straightforward mapping between the input rotation rate and the resulting thrust. This mapping can be inverted to determine the rotation rate needed to generate a desired thrust. However, in complex systems, such as flapping-wing robots where intricate fluid motions are involved, mapping inputs (wing kinematics) to outcomes (aerodynamic forces) is nontrivial and inverting this mapping for real-time control is computationally impractical. Here, we report a machine-learning solution[1] for the inverse mapping of a flapping-wing system based on data from an experimental system we have developed. Our model learns the input wing motion required to generate a desired aerodynamic force outcome. We used a sequence-to-sequence model tailored for time-series data and augmented it with a novel adaptive-spectrum layer that implements representation learning in the frequency domain. To train our model, we developed a flapping wing system that simultaneously measures the wing's aerodynamic force and its 3D motion using high-speed cameras. We demonstrate the performance of our system on an additional open-source dataset of a flapping wing in a different flow regime. Results show superior performance compared with more complex state-of-the-art transformer-based models, with 11% improvement on the test datasets median loss . Moreover, our model shows superior inference time, making it practical for onboard robotic control. Our open-source data and framework may improve modeling and real-time control of systems governed by complex dynamics, from biomimetic robots to biomedical devices.

## 1 Introduction

In machine learning frameworks that model causal relationships, *e.g.*, for prediction, causes are typically mapped to their effects. This forward mapping serves a wide range of applications, for example, predicting the motion of a mechanical system based on the forces acting on it (Dearden & Demiris, 2005), or in weather forecasting (Yu et al., 2024), in which data on previous atmospheric conditions are used to forecast future weather. In other cases, though, rather than predicting a system's response to a set of conditions or inputs, it is required to *control* the system by modulating its inputs to achieve a desired outcome. Hence, systems control would benefit from an *inverse* mapping, which flips the causal relationship by mapping the desired outcome to the input that would have led to this outcome. Such inverse mapping would enable the design of a controller that applies these inferred inputs to achieve desired behaviors. If the relationship between causes and their effects can be readily inverted, then an inverse-mapping control approach is useful and simple to implement. For example, in many robotic systems, a desired mechanical motion (outcome) can be directly mapped to the forces and torques that can generate it (cause) (Nguyen-Tuong & Peters, 2011).

---

*Webpage: https://www.beatus-lab.org
[1] Framework, models, and data are publicly available on github.com/Hadar933/AdaptiveSpectrumLayer

Here, in contrast, we address the case where the forward mapping between inputs and outcomes is nontrivial and difficult to calculate and invert. Examples of such systems include the mapping between pacemaker signals and a desired heart activity Simantirakis et al. (2009), and the mapping from injected insulin dosage to the resulting blood glucose level in a specific patient (Thomas & Heinemann, 2022). An additional noteworthy class of such systems is systems involving complex fluid motion, such as a flapping wing of an insect or a flapping-wing micro-air-vehicle (FW-MAV) (Dickinson et al., 1999; Sane & Dickinson, 2002; Tu et al., 2020; Bayiz & Cheng, 2021b; Keennon et al., 2012; Ma et al., 2013; Jafferis et al., 2019; Karásek et al., 2018; Coleman et al., 2015; Nguyen & Chan, 2018), illustrated in Fig. 1 and Supplementary Movie 1. The fluid dynamics in these systems are highly nonlinear and complex (Sane, 2003; Ellington et al., 1996): the wing induces intricate vortex structures that determine the aerodynamic forces; during flapping the wing interacts with its own, previously generated, flow field, which introduces complex time dependencies; and, finally, flapping wings often deform due to their elasticity and interaction with the flow, which then effects back on the flow itself, and so on, resulting in a complex fluid-structure interaction (Shyy et al., 2010; Nakata & Liu, 2012; Miller & Peskin, 2009; Young et al., 2009). Therefore, the forward mapping from wing motion (cause) to the aerodynamic force (outcome) often requires either using a mechanical, scaled-up flapping-wing analog (Dickinson et al., 1999; Bayiz & Cheng, 2021b; Whitney & Wood, 2010; Ellington et al., 1996; Muijres et al., 2014; Hsu et al., 2019; Melis et al., 2024), or numerical solution of the Navier-Stokes flow equation, which is highly computationally intensive and impractical for online system control. Quasi-steady-state approximations of the aerodynamic force are available and relatively simple to invert (Dickinson et al., 1999; Sane & Dickinson, 2002; Whitney & Wood, 2010; Nakata et al., 2015), however, they might become less accurate on sub-wingbeat resolution and are, hence, typically used for evaluating wingbeat-averaged forces (Bomphrey et al., 2017; Dickinson et al., 1999; Brunton et al., 2013).

Current FW-MAV designs circumvent the complexity of fluid dynamics by using a set of single-axis linear controllers based on insect-inspired control heuristics with manually-tuned parameters (Keennon et al., 2012; Ma et al., 2013; Coleman et al., 2015; Karásek et al., 2018; Nguyen & Chan, 2018; Jafferis et al., 2019; Tu et al., 2020). Although this simplified approach has made stable flight and maneuvers possible, it might be sub-optimal in allowing these vehicles to exploit their full performance envelope and achieve the remarkable agility and robustness of flying insects and hummingbirds. The aerodynamics of a flapping wing is, therefore, an appealing and practical test-bed for modeling the inverse mapping of a complex system. In this work, we address this complex inverse-mapping problem using deep-learning tools. To the best of our knowledge, there has been limited exploration of this approach in the existing literature.

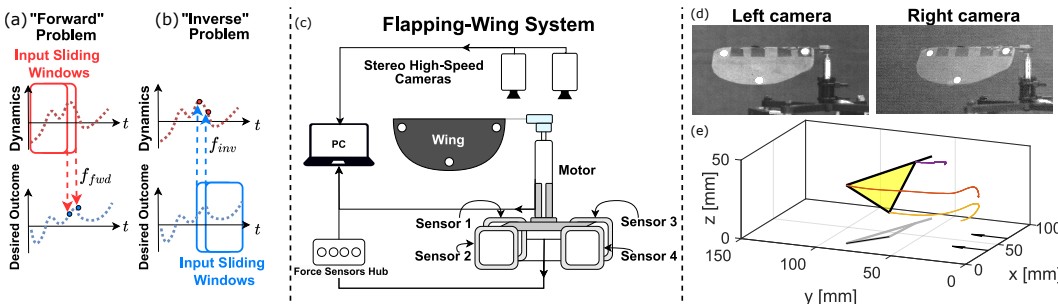

Figure 1: **Forward vs. inverse mapping of a physical system.** $(a)$ In forward mapping, a model $f_{\text{fwd}}(t)$ predicts system outcomes based on the tracked system dynamics, *e.g.*. Given the history of the wing motion, predicting the current lift force generated by the wing. $(b)$ In inverse mapping, a model $f_{\text{inv}}(t)$ takes in future/desired system outcomes to infer the inputs that generate them. For the wing, using the future lift force to predict what wing motion created this force. $(c)$ A diagram of a wing driven by a motor, with force and camera sensors. (d) Experimental setup: sample images from the two fast cameras, showing the wing and its markers. (e) The 3D position of the wing in motion. The yellow triangle represents the triangulation of the three markers and colored lines indicate the markers' trajectories. Two black arrows show the cameras' viewpoint.

Here, we present a machine-learning model for the inverse mapping of experimental flapping-wing systems that learns the input wing motion required to generate a desired aerodynamic force outcome. We use a sequence-to-sequence model (Bahdanau et al., 2014) that we tailored for time-series data. Our framework employs a bidirectional recurrent neural network (RNN) backbone (Schuster & Paliwal, 1997) combined with a time-attention mechanism and an Adaptive Spectrum Layer (ASL), which uses both amplitude and phase information to capture the intricate dependencies between motion and resulting forces in both the time and frequency domains. To generate a dataset of wing kinematics and aerodynamic forces for training and testing our model, we developed a mechanical flapping-wing system. Our system controls wing motion, measures it in 3D using two fast cameras, and simultaneously measures the forces generated by the wing. We trained and tested our model also on an open-source dataset of a flapping wing operating in a different flow regime (Bayiz & Cheng, 2021b;a). Our approach demonstrates comparable and even superior performance when compared with state-of-the-art transformer-based models (Vaswani et al., 2023), with 11% improvement on the test datasets. The ASL performs representation learning in Fourier space, allowing for the mitigation of noise and amplification of important frequencies, which becomes especially helpful in analyzing periodic systems such as flapping wings. Learning such inverse-mapping problems may directly improve the control of systems governed by complex dynamics, such as fluid motion. In FW-MAVs, for example, integrating such a trained network into the flight controller would enable efficiently calculating the wing kinematics required for exerting desired forces and torques on the vehicle, thereby exploiting its full performance envelope. We believe that this framework can apply to other complex domains, from robotics to biomedical devices.

## 2  RELATED WORK

### 2.1  FORWARD-MAPPING MODELING

Several approaches have been used for forward mapping modeling of flapping wing systems, that is, finding the aerodynamic forces resulting from a given wing motion. One direct method is measuring the aerodynamic forces on a scaled-up wing model flapping in a fluid and mimicking the motion of, for example, experimentally measured kinematics of an insect's wing (Dickinson et al., 1999; Bayiz & Cheng, 2021b; Whitney & Wood, 2010; Ellington et al., 1996; Muijres et al., 2014; Hsu et al., 2019). With proper scaling of the wing motion and fluid viscosity, the forces measured on the scaled-up model can be rescaled back to the corresponding insect forces. Another method is Computational Fluid Dynamics (CFD), where the Navier-Stokes flow equation is numerically solved on a spatial grid and the aerodynamic forces on the wing are then calculated from the solved flow. While CFD has been instrumental in understanding the fluid dynamics of flapping wings (Dickinson & Muijres, 2016; Nakata et al., 2015), insect stability(Gao et al., 2011; Sun, 2014; Perl et al., 2023), and complex fluid-structure interactionsYoung et al. (2009); Shyy et al. (2010); Nakata & Liu (2012); Miller & Peskin (2009), this class of methods is computationally intensive and, hence, impractical for inverse modeling in a real-time flight controller. A dramatic simplification is offered by quasi-steady-state (QS) aerodynamic models, which approximate the aerodynamic force of a wing as a function of its instantaneous motion Dickinson et al. (1999); Sane & Dickinson (2002); Weis-Fogh (1973); Whitney & Wood (2010). For a specific wing geometry, QS models can be calibrated and tuned based on a scaled-up mechanical wing model Dickinson et al. (1999); Whitney & Wood (2010) or CFD simulationsNakata et al. (2015). Because these models provide an analytical form of the aerodynamic force, they can, in principle, be inverted and used for real-time control. Yet, QS models neglect complex flow-related features, for example, wing vorticity, wing interaction with its previously generated flow, and fluid-structure interaction, which may be important for utilizing the full capabilities of FW-MAV.

Deep learning models have been applied for problems in fluid dynamics, such as turbulence and flow controlLing et al. (2016); Brunton et al. (2016); Duraisamy et al. (2019), and for mitigating windy conditions and structural damage in quad-copter control O'Connell et al. (2022). The impressive achievements in quad-copter control do not require significant inverse modeling due to the relatively simple mapping between desired forces and torques and rotor speed. This is markedly different than the complex aerodynamics of flapping wings. For a flapping wing, deep learning models enable capturing the complex forward mapping from wing motion to aerodynamic forces without simplifying assumptions. In this approach, a model is trained on a dataset of measured or calculated forces obtained from a scaled-up mechanical model or CFD, respectively, based on a set of predefined

wing kinematics. A trained model can potentially predict these forces for given input kinematics and do so much faster than CFD models and potentially more accurately than QS models. The applicability of such a model depends, naturally, on the quality and breadth of the dataset it has been trained on. In 2021, Bayiz and Cheng introduced a state-space deep learning approach that accurately predicted aerodynamic forces using data from a scaled-up mechanical wing Bayiz & Cheng (2021b). Their model, trained and tested with 548 diverse wing kinematics, demonstrated the predictability of aerodynamic forces based on a half-wingbeat look-back window of the previous wing kinematics. Here, we invert this dataset to model the system's inverse aerodynamics.

Beyond aerodynamics, forward mapping modeling applies to simulating diverse mechanical systems, such as humanoid robot motion Tassa et al. (2012). In these simulations, the state of the system encompasses generalized positions and velocities, governed by equations of motion incorporating inertia tensors, external forces, and control inputs. Using a semi-implicit Euler integration scheme enables the iterative calculation of the system's state based on applied actions.

## 2.2 TIME SERIES MODELING AND FOURIER ANALYSIS

Time-series modeling has been fundamental across various disciplines, including climate modeling Mudelsee (2019), biological sciences Stoffer & Ombao (2012), and finance Böse et al. (2017). Traditional methods such as auto-regressive and exponential smoothing rely on domain expertise Box et al. (2015), but modern machine learning techniques are increasingly adopted owing to their data-driven nature and scalability Lim & Zohren (2021). Recent architectures that use FFT (Fast Fourier Transform) include the following: AutoFormer features a distinct architecture: the encoder emphasizes modeling the data periodicity, while the decoder includes accumulation structures for trend-cyclical components and stacked auto-correlation mechanisms for periodic components. This mechanism replaces traditional self-attention methods and efficiently computes auto-correlation using FFT Wu et al. (2021). Similarly, FedFormer introduces low-rank approximated transformation in the frequency domain to expedite attention mechanisms in time series forecasting Zhou et al. (2022). Adaptive Temporal-Frequency Networks utilize FFT to extract trend and periodic features for improved forecasting accuracy Yang et al. (2022). StemGNN employs Graph Fourier Transform (GFT) and Fourier transform to capture inter-series correlations and temporal dependencies effectively Cao et al. (2020). Notably, StemGNN automatized the learning of inter-series correlations from data, leveraging spectral representations for prediction.

In Fourier Neural Operator (FNO), a parameterized low-pass filter in Fourier space facilitates the learning of mappings from functional parametric dependencies to solutions, thus enabling the exploration of a broad family of partial differential equations Li et al. (2020). Additionally, the random Fourier method and random Fourier softmax (RF-softmax) technique offer efficient and accurate random sampling, exploiting frequency-space features Rawat et al. (2019).

The adaptive spectrum layer (ASL) reported here uses both the magnitude and phase of the Fourier spectra for representation learning, weighs frequency bins accordingly, and functions as a standalone representation layer. In contrast, other approaches often rely on FFT for efficient computation in Fourier space, as well as overlook the comprehensive information provided by the Fourier transform, or lack a gated weighing mechanism that considers all information from other frequency bins.

## 3 METHODS

We developed a flapping wing system (Fig. 1c) in which we measured the wing kinematics using high-speed cameras. The wing degrees of freedom were characterized by $M_K=3$ Euler angles (Fig. 2): the stroke angle $\phi$, elevation angle $\theta$, and wing pitch angle $\psi$. The aerodynamic forces were measured in sync with the wing kinematics using $M_F=4$ vertical force sensors. The essence of inverse-mapping modeling in this system lies in learning the relationship between desired output aerodynamic forces and the input wing kinematics that generate them. Intuitively, the input to the system is a time sequence of desired aerodynamic forces that the wing should generate. The output is the full wing kinematics that, when applied to the wing, would result in the desired forces. The system learns the inverse mapping in two parallel backbones: learning the time-dependent (Seq2Seq) and frequency-dependent (ASL) relationships between the forces. As a result, both the timestamps and frequencies that are relevant for the mapping are learned.

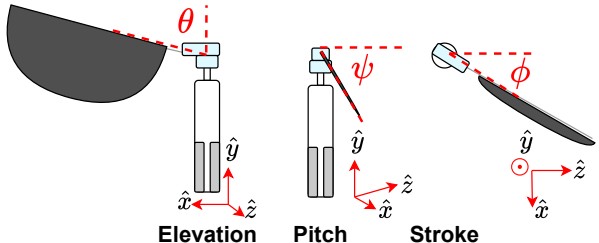

Figure 2: **Wing degrees-of-freedom.** The three angles of wing rotation: elevation angle $\theta$ (left, shown in a side view), wing-pitch angle $\psi$ (middle, shown in a front view), and wing-stroke angle $\phi$ (right, shown in a top view). The $\hat{x}$, $\hat{y}$, $\hat{z}$ vectors represent the Cartesian lab frame of reference.

## 3.1 MULTI-VARIATE MULTI-TARGET TIME SERIES FRAMEWORK

**Problem Definition**

We reformat the generalized time series formalism to our inverse-mapping modeling framework of {desired future outcomes}→{required input kinematics}. Without loss of generality, we consider wing kinematics (Fig. 2) as required system input, and system-measured forces, provided from a set of sensors (Fig 1c) as desired future outcomes. given $N$ distinct time-series events that represent data from $M_F$ (force) sensors $\left\{ \{F_{1:t_0}^{i,j}\}_{j=1}^{M_F} \right\}_{i=1}^{N}$ where $F_{1:t_0}^{i,j} \in \mathbb{R}^{t_0}$ indicates the force values of the $j$'th sensor in the $i$'th dataset at times $1, 2, ..., t_0$. Our goal is to predict the corresponding $N$ distinct wing kinematics $\left\{ \{K_{1:t_0}^{i,j}\}_{j=1}^{M_k} \right\}_{i=1}^{N}$ where $K_{1:t_0}^{i,j} \in \mathbb{R}^{t_0}$, indicates the kinematic values of the $j$'th degree of freedom in the $i$'th event at times $1, 2, ..., t_0$. To simplify our notation, we will mostly describe the kinematics in terms of three angles ($M_K{=}3$) $\{\phi_{0:t_0}^i, \theta_{0:t_0}^i \psi_{0:t_0}^i\}_{i=1}^N$, where $\phi_t^i, \theta_t^i, \psi_t^i \in \mathbb{R}$ represent the stroke, elevation, and pitch angle at time $t$ in the $i$'th event, respectively (Fig. 2). Formally, we aim to model the following conditional probability distribution:

$$p\left(\phi_{t=0:t_0}^i, \theta_{t=0:t_0}^i \psi_{t=0:t_0}^i \middle| \{F_{t_0:t_0+\tau}^{i,j}\}_{i,j=1}^{M_F}; \Phi \right) \tag{1}$$

This is the probability of wing kinematics at times $t{\in}[0, t_0]$ given the set of future forces at times $t'{\in}[t_0, t_0 + \tau - 1]$ at a fixed future window of size $\tau$ that they generated. These, alongside the learnable parameters $\Phi$, are optimized to minimize loss using an SGD-like process. In practice, we reduce the problem to learning a one-step-ahead prediction model, for any $i{\in}[N]$

$$p\left(\phi_t^i, \theta_t^i, \psi_t^i \middle| \{F_{t:t+\tau}^{i,j}\}_{j=1}^{M_F}; \Phi \right), \tag{2}$$

where an optimal model $f$ for which $\phi_t^i, \theta_t^i, \psi_t^i \sim f\left( \{F_{t:t+\tau}^{i,j}\}_{j=1}^{M_F} \right)$ is explored, to predict the distribution of wing kinematics at time $t$ given the measured forces at time $t$.

## 3.2 SEQUENCE-TO-SEQUENCE

We instantiate our model $f$ as a Seq2Seq model Bahdanau et al. (2014), adjusted for time series. More precisely, we utilize an RNN (Recurrent Neural Network) encoder embedding alongside an RNN decoder intertwined with an attention mechanism to predict the next time step given all previous window values (Fig. 3). The encoder transforms the input time series data into an embedded representation using an Adaptive Spectrum Layer (see below), followed by a bidirectional GRU (Gater Residual Network). The final hidden states from both forward and backward passes are concatenated and reshaped through a linear layer to serve as the initial state for the decoder. Crucially, an attention mechanism computes attention weights based on the decoder's current hidden state and all the encoder's outputs. This ensures the decoder focuses on relevant time steps from the encoder when predicting subsequent values. The decoder starts with the last value of the input sequence and, guided by the attention mechanism, iteratively generates predictions for the forecast horizon.

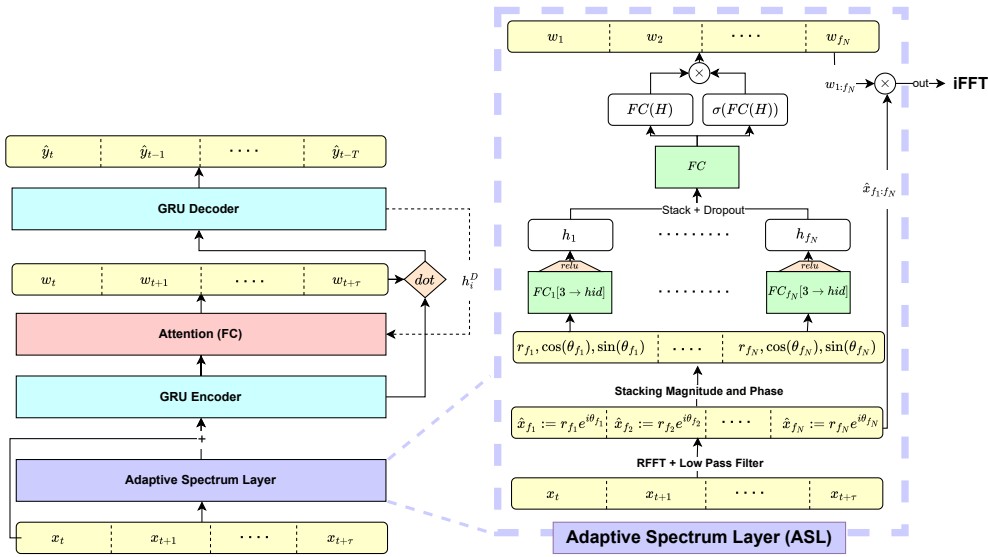

Figure 3: **System architecture: Seq2Seq with ASL.** The input sequence $x$ is encoded by an adaptive spectrum layer (ASL). ASL conducts representation learning in Fourier space, assigning weights to each frequency bin using the entire complex signal, and then reverting to the time domain via IFFT. A skip connection is added from input to representation. Subsequently, a GRU encoder generates a fixed-size representation. Following this, the attention (fully connected, FC) mechanism utilizes the current decoder hidden state and encoder context vector to compute attention weights $w_t$. The last encoder state is employed instead of the (non-existent) decoder state in the initial iteration. These attention weights adjust the encoder context vector based on the current decoder hidden state. Finally, the resulting weighted tensor passes through a GRU-based decoder to predict the next step $\hat{y}_t$, with $T$ representing the prediction window size.

**Adaptive Spectrum Layer.** Motivated by the potential benefits of frequency domain analysis, we introduced the Adaptive Spectrum Layer (ASL) layer, shown in Fig. 3. The ASL takes in the raw input signal and applies representation learning in Fourier space. It then combines the new representation and the signal via a skip connection and propagates this output to the intermediate layers of the neural network in which it is encapsulated.

The forward pass begins by applying a real-valued (symmetric) Fourier transform (FFT) on the raw input signal $x$. Then, a low-pass filter is used to retain relevant frequency information below a set frequency. The magnitudes $r_{f_i}$ and phases $\theta_{f_i}$ of each Fourier component (the $i$'th bin) are then concatenated to form a new complex tensor $\hat{x}$, in which every entry consists of the respective magnitude and the phase represented by its sine and cosine values. We then use a fully connected layer (FC) with additional non-linearity to derive a new representation from $\hat{x}$. Subsequently, a dropout layer is applied to the stacked hidden representation $h_i, ..., h_{f_N}$, after which a fully connected layer transforms the representation to a bounded weight vector $w_1, ..., w_{f_N}$ using a simple gating mechanism. In this vector, every entry corresponds to the weight associated with its respective frequency bin in Fourier space. Lastly, the learned weights are point-wise multiplied with the original complex vector, and an inverse Fourier transform (IFFT) is used to convert the new signal back to the time domain.

## 4 DATASETS

### 4.1 OUR MEASURED DATASET: A WING FLAPPING IN AIR

**A Flapping Wing System.** We developed a flapping wing system (Fig. 1c-e, Supplementary Movie 1) that can move with a predetermined stroke angle profile $\phi(t)$, and with passively-determined elevation $\theta(t)$ and pitch $\psi(t)$ angles. That is, $\theta(t)$ and $\psi(t)$ are outcomes of the dynamic interactions between the wing and the surrounding air, wing inertia, and wing elasticity Beatus & Cohen (2015).

The wing consisted of a $12.7\mu$m thick Mylar sheet, with a leading edge made of a carbon-fiber rod 1mm in diameter. Wing span was 7cm and its maximum chord was 3cm. A 3D-printed hinge connected the wind to a brushless DC motor (Maxon ECXSP06M BL KL-A-HP-12V) with a built-in 15:1 gear ratio and an angular position encoder. Thus, the motor was driving the wing directly by following a predetermined stroke angle profile $\phi(t)$ using a designated controller we developed.

**Measurement Setup.** The measurement setup consisted of two modules: a fast-imaging setup that measured wing kinematics (input), and force sensors that measured the vertical force generated by the wing (outcome). The imaging setup included two high-speed cameras (Phantom v2012, Vision Research) in a stereo configuration (Fig. 1c-e, Supplementary Movie 1) with parallel optical axes. The cameras were mutually calibrated to find their intrinsic and extrinsic matrices. In each experiment, the two cameras recorded the flapping wing and operated in sync at 10,000 frames per second. To track the wing, we attached three white circular markers onto the wing's camera-facing surface and tracked the markers' positions in each camera view using basic segmentation and optical flowHorn & Schunck (1981). Subsequently, we used the 2D markers' trajectories from both cameras to triangulate the position of the markers in 3D in the lab frame of reference based on the cameras' calibration. Finally, under the verified assumption that the wing maintained a flat shape during its motion, we converted the three 3D marker coordinates into the standard Euler angle description for flapping wings of $\phi(t), \theta(t), \psi(t)$.

The second experimental module – for force measurement – consisted of four force sensors (SI-USB4, Interface Inc.) arranged in a symmetric cross configuration, with the motor attached at its center (Fig. 1c-e). The sensors measured in sync at 5,000 samples per second, and their four readouts of vertical forces represent the aerodynamic force output generated by the wing. Treating all readouts as separate signals, rather than, for example, summing them, is relevant for torque calculation and phase-related feature extraction. Finally, the two data streams of the wing angles and forces were synchronized and combined to form a single event in our dataset.

**Data Collection.** We measured a total of $N{=}153$ events that span different wingbeat frequencies and $\phi(t)$ profiles. This dataset is publicly available.Each event corresponds to a wing trajectory and force data, $2{-}6$ wing-beats, or $0.11{-}0.75$ seconds, long. The range of profiles was obtained by using a parameterization for $\phi(t)$ Berman & Wang (2007) that provides a continuum of profiles from sinusoidal to triangular as a function of a single parameter $K{\in}[0,1]$:

$$\phi(t) = \frac{1}{2}\Phi \frac{\sin^{-1}\left(K\sin(2\pi ft)\right)}{\sin^{-1}K}, \tag{3}$$

where $f$ is the wingbeat frequency and $\Phi$ is the stoke peak-to-peak amplitude. Our dataset included events with $f{\in}[0,20]$Hz and $\Phi{\in}[\pi/6, \pi/3]$rad. Due to the range of wing speeds in the dataset, the wing's Reynolds number was $1,000{-}50,000$, which covers the flow regimes of medium to large insects, small birds, and FW-MAVs.

The resulting dataset (Table 1, Fig. 4) is formulated as $\mathcal{D} = \left\{\{F^{i,f}_{1:t_{0_i}}\}^{M_F}_{f=1}; \{K^{i,k}_{1:t_{0_i}}\}^{M_K}_{k=1}\right\}^N_{i=1}$. For each event $i{=}1, 2, ..., N$ there are $M_F$ force signals with $t_0$ entries and $M_K$ kinematic signals with $t_0$ entries. We use $M_F{=}4$ (four force sensors) and $M_K = 3$ for stroke, pitch, and elevation angles. The upper scripts $f \in [M_F]$ and $k \in [M_K]$ represents the $f$'th force sensor and $k$'th kinematics respectively. Also, note that $t_{0_i}$ is not the same for all $i$, allowing events of different duration.

| Dataset | No. of Events | Samples per event | Input dim. | Output dim. | Total samples | Sample Rate |
|---|---|---|---|---|---|---|
| Our | 153 | 550–3787 | 4 | 3 | 438,552 | 5,000 Hz |
| Open Source | 548 | 480 | 5 | 3 | 263,040 | 25 Hz |

Table 1: **Datasets.** The two datasets used in the comparative analysis. "Our" dataset is the experimental flapping-wing dataset measured in our system, and "Open Source" represents the flapping-wing dataset reported in Bayiz & Cheng (2021a). Each event represents an individual wing kinematic profile. Event duration is counted in samples.

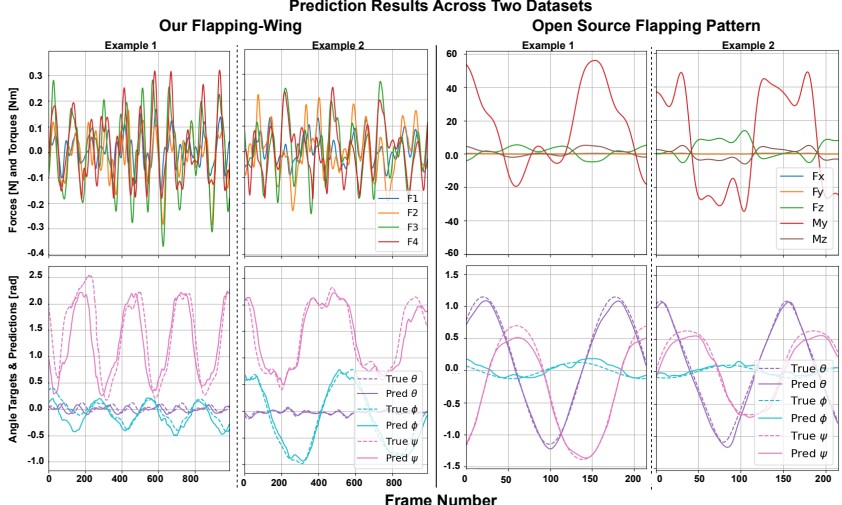

Figure 4: **Prediction examples.** Four pairs of input-output scenarios from our dataset (left, described in 4.1) and the open source dataset Bayiz & Cheng (2021a) (right, described in 4.2). The upper section displays force/torque inputs representing the desired system outcome. In our dataset, these are $F_1, F_2, F_3, F_4$ as depicted in our experimental setup (see Fig. 1c-e). In the open source dataset, the outcome is represented by a set of three measured forces $F_x, F_y$, and $F_z$, and two measured torques $M_y, M_z$. In both experiments, the targets are similar and represented in the lower section as the corresponding true angle labels and predicted angles, generated by our adapted Seq2Seq+ASL model trained to model the inverse mapping. Different system outcomes (top) result from different system dynamics (bottom) in each event. The events shown span various wing kinematics.

## 4.2 OPEN SOURCE DATASET: A WING FLAPPING IN VISCOUS FLUID

The second dataset we used (Table 1, Fig. 4) has been published by Bayiz & Cheng (2021b), who measured the wing kinematics and aerodynamic forces of a plate-like wing flapping in mineral oil. Wing kinematics $\{\phi, \theta, \psi\}$ was controlled by step-motors and the aerodynamic forces were measured by three forces- and two torque-sensors. The Reynolds number of the system was ∼1000, modeling medium-sized insects. This dataset included 548 events with various kinematics, each having 480 samples of synchronized kinematics-force data taken at 25Hz. Bayiz and Cheng used this dataset to develop a deep-learning system that learned the forward-mapping: from wing kinematics to forces and torques. Here, we used this dataset in reverse to learn the inverse mapping.

## 5 RESULTS

We trained an individual model on the two datasets, to infer the wing kinematics that generated a given force time-series. The input signals in the two datasets have different dimensionality, units (force vs. force and torque), and sampling rates. Both datasets were randomly divided such that 75% of the events were used as a training set, 10% reserved for validation, and 15% for testing. Throughout the training process, a standard hyper-parameter tuning loop was used to search through hyperparameter space including model parameters (*e.g.*, the number of attention heads, and the size of the hidden representation vector) and input window characteristics (history size, batch size, learning rate, normalization schemes, *etc*.). For further technical details see Appendix A.3.

Model performance was evaluated on the test set using the Mean Absolute Error (MAE) loss of the predicted wing angles. The MAE for each angle was calculated per event, averaged across the three predicted angles per event, and then averaged across each test set. The resulting model architectures were relatively small, with a few hundred to ∼200,000 parameters. Hence, training one model took ≤3min on a single Nvidia RTX 3090 GPU with 24 GB RAM, and an exhaustive hyperparameter search took ∼50hr on the same hardware.

| Methods | | Linear | AutoFormer | FedFormer | Seq2Seq+ASL | Seq2Seq | Transformer | NLinear |
|---|---|---|---|---|---|---|---|---|
| Ours | Mean | 0.2662 | 0.2380 | 0.3049 | **0.1323** | 0.1471 | 0.1476 | 0.2800 |
| | Median | 0.2643 | 0.2271 | 0.2920 | **0.1216** | 0.1236 | 0.1430 | 0.2836 |
| OpenSrc | Mean | 0.3223 | 0.3626 | 0.3003 | 0.1130 | 0.1206 | **0.1123** | 0.3258 |
| | Median | 0.3070 | 0.3379 | 0.2875 | **0.0908** | 0.1058 | 0.1021 | 0.3131 |

Table 2: **Results.** Comparison of Mean Absolute Error (MAE) in radians between various models on both Ours and the open source datasets, aggregated across all test events using Mean or Median. Best- and second-best-performing models are highlighted in **bold** and underscore, respectively.

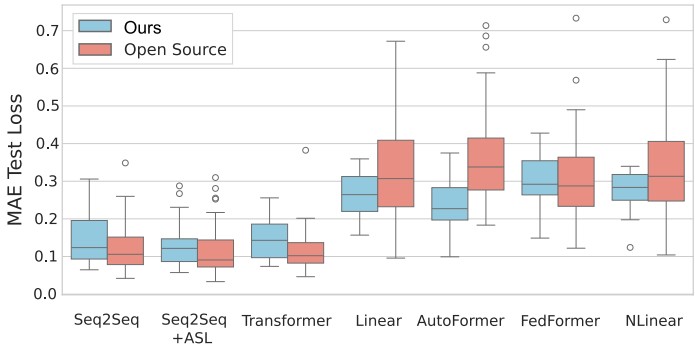

Figure 5: **Comparison with state-of-the-art models.** The distributions of test loss across seven models for two datasets: Our dataset and the open-source dataset Bayiz & Cheng (2021b). Inside each box, the horizontal line represents the median MAE, the colored box represents the 2nd and 3rd quartiles, and the whiskers represent the 1st and 4th quartiles. Outliers are indicated by open circles. adapted Seq2Seq+ASL model demonstrates superior performance, particularly evident in its median values, outperforming other models. Interestingly, Seq2Seq+ASL has more outliers than the Transformer, which explains the difference between their mean and median metrics.

The performance of our models, with and without ASL, on the two test sets are shown in Table 2 and Fig. 5, with examples of representative events in Fig. 4. Additionally, these results are compared with the performance of several state-of-the-art models on both datasets (implemented in Zeng et al. (2023)): Transformer Vaswani et al. (2023), AutoFormer Wu et al. (2021), FedFormer Zhou et al. (2022), NLinear Zeng et al. (2023), and a Linear model represented fully connected layer that is applied on the flattened input (see the Appendix for further details).

**Inference accuracy.** Our results show that the inverse dynamics problem of a flapping wing system can be well-approximated by deep learning models. First, the data in Table 2 shows that the ASL is consistently improving performance by 11.2%, 1.7%, 6.7% and 16.5% compared with Seq2Seq alone. The improvement offered by the ASL frequency-space representation layer demonstrates the benefit of using such representations, especially for periodic signals. Second, Seq2Seq+ASL is consistently the first- or second-best model within the tested models and metrics. On our dataset with MAE metrics, Seq2Seq+ASL outperforms the Transformer on $2/3$ of the test set (Wilcoxon signed rank test $p_{\text{value}}=0.06$) and is the best-tested model, improving by 10% concerning standard Seq2Seq and with 10.4% compared with Transformer. On the same dataset but using the median metrics, Seq2Seq+ASL, and standard Seq2Seq show equivalently best performance, improving by 13.6% compared with Transformer. On the Open Source dataset Seq2Seq+ASL and Transformer perform equally well under the MAE metric, (Seq2Seq+ASL outperforms the Transformer in 54% of the test set, Wilcoxon signed rank test $p_{\text{value}}=0.58$) . Under the median-MAE metric, Seq2Seq+ASL is the best-tested model, improving Transformer by 11%. In most tested cases, the performance of the Seq2Seq+ASL model suggested here is superior to more sophisticated state-of-the-art models.

**Inference time.** even though Seq2Seq+ASL has $\times 4$ more parameters ($\sim$200k) compared with the Transformer model ($\sim$50k, Appendix), The inference time of Seq2Seq+ASL is $\times 10$ shorter

compared with Transformer ($2.00\pm0.13$ms vs. $19.53\pm5$ms). This difference makes Seq2Seq+ASL more practical for integration in onboard flight controllers of FW-MAVs, while the transformer latency is too long with respect to the typical wingbeat period of the existing prototypes. The scaling of the inference time with the number of parameters of each of these two models shows that Seq2Seq+ASL inference time is practically constant with the number of parameters, while the Transformer inference time increases significantly with the number of parameters (Appendix)

**Ablation tests.**    To characterize the functionality of the ASL, we performed a suite of ablation tests on all of its building blocks. We tested the sine/cosine vs. angular representation, low-pass filter cutoff frequency, ASL gating mechanism, ASL per-frequency layer (combining different frequency bins after FFT or not), and learning a new complex number representation from the FFT vector or not. The full results are given in Supplementary Table 1. Briefly, encoding the phase data as sine/cosine was slightly, but not conclusively better than an angular representation (probably due to the improved encoding of periodicity of sine/cosine). Other attributes, such as the low-pass filter cutoff frequency, were highly significant. Testing three different frequencies: 20, 100, and 210Hz, we see that all of the top-10 models had a cutoff frequency $\geq$100Hz.

# 6    CONCLUSION

We presented an inverse-mapping modeling framework, in which the required system inputs (wing kinematics) are predicted given the desired system outcomes (aerodynamic forces). A specific realization of such a model was explored using an experimental flapping wing system from which we collected a dataset capturing the relationship between wing kinematics and the resulting forces. The task was then formulated as an inverse mapping from the output forces of the system to the wing kinematics that generate them. To model this problem, we proposed a deep learning architecture based on a sequence-to-sequence (Seq2Seq) model with an Adaptive Spectrum Layer (ASL). The ASL performs representation learning in the Fourier domain, using both amplitude and phase information. As such, the ASL captures important frequency patterns, which is particularly beneficial for periodic signals like those in flapping wing systems, and filters out undesired noise. On both datasets, which represent two flow regimes, our model demonstrated superior performance compared with other state-of-the-art models, achieving up to $\sim$11% improvement. These results support the current view that Transformer-based models are not necessarily optimal for time-series analysis Zeng et al. (2023). Further, our RNN implementation is expected to be significantly more computationally efficient than Transformer models, which would benefit deployment in onboard systems.

Interestingly, on the current task, our model outperformed FedFormer, which also uses frequency data. This may be explained by the fact that FedFormer is a forecasting model in which the input and output dimensions are identical, unlike in our use case. To apply FedFormer to such cases, one had to change its output dimension, which might have hindered its performance. More crucially, to implement an attention mechanism in linear-time complexity, FedFormer implements random frequency sampling on which it builds its frequency-domain representation. Yet, in our case, because we did not use computationally-expensive attention, we could use a frequency-domain representation on the entire spectrum, without data loss.

**Limitations.**    First, due to the challenges of developing a fully operational flapping robotic device, our model was demonstrated on bench-top device with a single wing. The model could be incorporated into FW-MAVs by training in a similar bench-top configuration and then including it as a module in the onboard controller that converts force and torque commands into wing actuation. Second, due to the limits of our datasets, our system was not trained to generalize on arbitrary wing forms and Reynolds numbers. Further, the wing-hinge in our system has a single axis, which might limit the repertoire of wing motions, compared, for example, with better-controlled wing models. Yet, this simple actuation may, in fact, fit with current FW-MAV designs, since all of them use a similar single-axis drive, relying on fluid-structure interaction to yield complex insect-like wing kinematics.

In summary, despite these limitations, we believe that the proposed inverse-mapping framework could be seamlessly integrated into a wide range of applications and improve the modeling and control of complex systems, from biomimetic robots to biomedical devices.

*This work was supported by the Israel Ministry of Science and Technology Grant No. 3-17400.*

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

# A APPENDIX

## A.1 ARCHITECTURE

For completeness, the full architecture is provided in Fig. 6

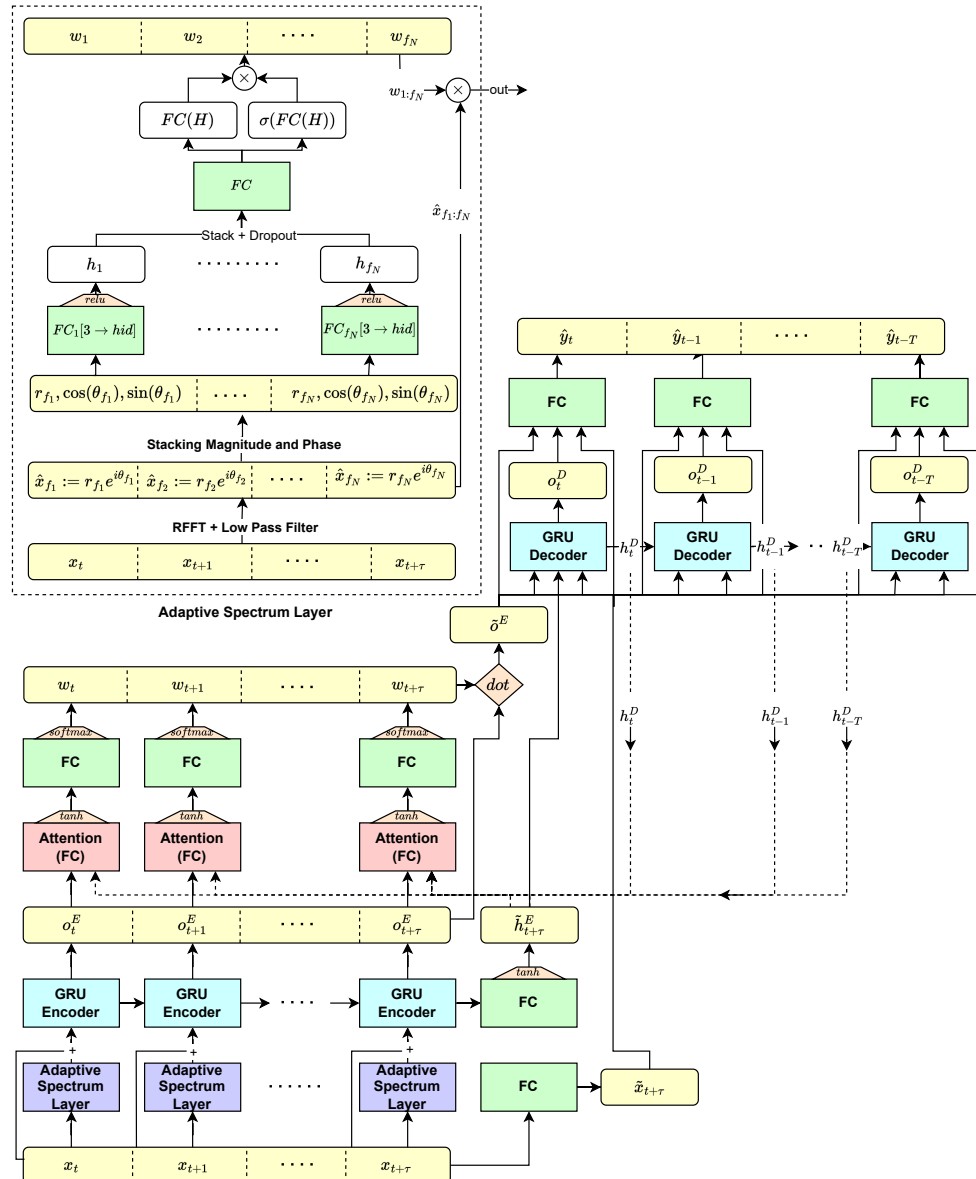

Figure 6: **Seq2Seq with ASL: Full Architecture.** Complementary to the high-level description in Fig. 3, the complete description of the framework is shown. The ASL layer (Purple), is also provided in detail in the top right. In the full architecture, the propagation of the hidden decoder states and the inner workings of the simple attention mechanism are also shown.

## A.2 ADAPTIVE SPECTRUM LAYER (ASL)

We extended the Seq2Seq model with ASL, a layer that performs representation learning of the input signal in the frequency domain, enhancing the model's ability to capture the underlying dynamics (Section 3.2), The ASL algorithm is described by the following pseudo-code:

---

**Algorithm 1** Adaptive Spectrum Layer (ASL)

---

**Require:** Tensor $x$ with shape $[H, F]$ (or batched with shape $[B, H, F]$),
**Require:** $f_{max}$, the maximal frequency to consider, and $f_s$ - the sampling frequency
**Require:** `dropout` rate $p \in (0, 1)$
  $N_f \leftarrow [0, 1, 2, ..., H/2 - 1, H/2] / (H/f_s)$
  $\hat{x} \leftarrow \texttt{rfft}(x) [:, : N_f, :]$                         ▷ Real valued Fast Fourier Transform
  $\hat{s} \leftarrow [|\hat{x}|, \cos(\angle\hat{x}), \sin(\angle\hat{x})]$             ▷ Stacking phase and magnitude
  $H \leftarrow \texttt{ReLU(FullyConnected}(\hat{s}))$       ▷ Hidden representation in Fourier space
  $w \leftarrow$ `Sequential(`
     • $H \leftarrow$ `dropout[p=`$p$`]`$(H)$
     • $H \leftarrow \texttt{FullyConnected}(H)$
     • $H \leftarrow H \times \texttt{sigmoid}(H)$
     • $H \leftarrow \texttt{sigmoid}(H)$
  `)`                                            ▷ Gating Mechanism
  $\hat{x} \leftarrow \texttt{Padding}(\hat{x} \times w)$ with $H - N_f$ zeros      ▷ Results in reconstruction of the same shape
  $x \leftarrow \texttt{irfft}(\hat{x})$

---

## A.3 HYPER-PARAMETER TUNING

In our framework, we offer the flexibility to explore a larger set of training-related hyperparameters, as specified in Tab. 3 A.3

| Hyperparameter | Description |
|---|---|
| `train_percent` | Percentage of data for training |
| `val_percent` | Percentage of data for validation |
| `feature_win` | Size of input window for temporal span |
| `intersect` | Intersection between feature and target window |
| `batch_size` | Number of samples in each mini-batch |
| `model_args` | Model-specific architecture arguments |
| `optimizer_name` | Choice of PyTorch optimizer |
| `criterion_name` | Loss and regularization criterion |
| `patience` | Epochs without validation improvement before stopping |
| `patience_tolerance` | Threshold for validation loss improvement |
| `n_epochs` | Total number of training epochs |
| `features_norm_method` | Normalization scheme for features |
| `targets_norm_method` | Normalization scheme for target values |
| `features_global_normalizer` | Global or per-dataset features normalization |
| `targets_global_normalizer` | Global or per-dataset target values normalization |
| `regularization_factor` | Regularization factor ($\lambda$) |

Table 3: **Training Hyperparameters.** Various training parameters offered as part of a training process

These hyperparameters collectively contribute to optimizing our model during the training process. In our experimental setup, we conducted an exhaustive search, particularly focusing on the hyperparameters of the Seq2Seq model, but not only. The parameters explored during this search included:

- **Seq2Seq model args:**
  - `embedding_size`: Varied from 5 to 50 to assess its impact on feature representation.
  - `hidden_size`: Varied to explore the influence of different hidden dimensions.
  - `n_layers`: Investigated different numbers of RNN layers (1-3 layers).
  - `attn_heads`: Explored different numbers of attention heads (1-10).
- **batch size**: Explored values ranging from a few dozen to a few thousand.
- **feature window size**: Varied (128-512) to assess sensitivity to temporal span.
- **normalizer schemes**: every possible pair of normalization schemes for the features and the targets, namely, every pair in {z-score, min-max, none} $\times$ {z-score, min-max, none}

With the new Adaptive Spectrum Layer (ASL, sec. 3.2), we explored the following hyperparameters:

- `hidden_size`: determines the size of the magnitude and phase representation learned by the ASL. It is typically set to a value similar in size to the hidden size of the Seq2Seq model, for convenience purposes.
- `dropout_rate`: Dropout is applied to the stacked hidden representations in the ASL to prevent overfitting. The dropout rate is a hyperparameter that controls the proportion of units to drop during training. We explored values ranging from 0.05 to 0.15, with 0.1 (10%) dropout being the most commonly used value.
- `use_freqs`: controls whether the static frequencies are concatenated to the input features. We initially explored using static frequencies but eventually decided not to use them as they did not change per window, which would introduce a constant bias.
- `cross_spectral_density`: represents an additional layer that computes the mean value of the Fast Fourier Transform (FFT) of the cross-correlation between force features. While this approach seemed promising, we ultimately did not use it in our final model.
- `frequency_threshold`: sets the maximum frequency to consider, acting as a low pass filter. We typically did not surpass 200Hz, and the choice of this threshold should be determined by observing the data and understanding its overall underlying frequencies, which can provide an inductive bias for our system.
- `Complexify`: This boolean hyperparameter controls whether a new complex vector is learned from scratch. If set to true, two designated layers are used to map the hidden representation to new magnitude and phase values. However, after experimentation, we decided not to use this addition as it did not improve our results and introduced unnecessary complexity to the model.

Concretely, the best-performing parameters are specified in Tab. 4 Furthermore, each model we evaluated has gone through similar hyperparameter tuning. Concretely:

**Transformer/AutoFormer Model Arguments**

- `d_model`: Dimension of the model, varied from 2 to 64 to assess the impact on feature representation capacity
- `n_heads`: Number of attention heads explored from 1 to 16
- `d_ff`: Dimension of feed-forward network, typically set to 2-64
- `e_layers, d_layers`: The number of encoder and decoder layers, varied from 1 to 2
- `moving_avg`: Kernel size explored in range 4-96
- `activation`: GeLU

**FEDFormer Model Arguments**

- `modes`: Selection method set to random
- `version`: Fixed to "fourier" as per model specification
- Other arguments: similar to Transformer

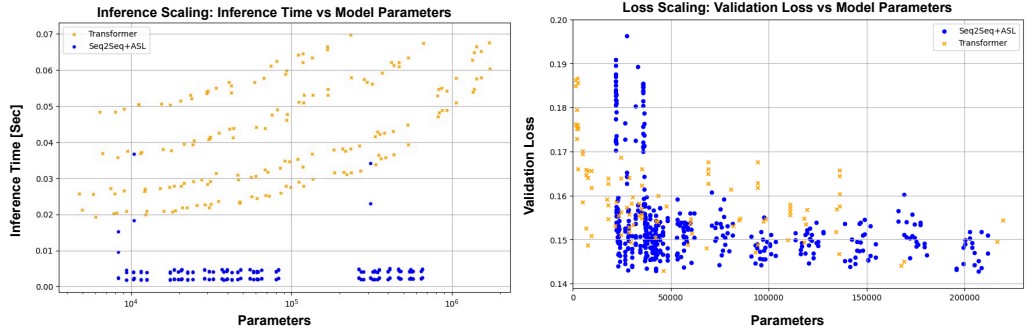

Figure 7: Scaling Analysis: (Left) Inference Time vs. Model Parameters; (Right) Validation Loss vs. Model Parameters; The Seq2Seq+ASL model demonstrates consistent improvements in MAE and significantly reduced inference time compared to the Transformer baseline.

**Informer Model Arguments**

- `factor`: ProbSparse attention factor tested in range [3, 5, 7]
- `distil`: Boolean parameter for encoder distilling mechanism
- Other arguments: similar to Transformer

**Linear/NLinear Model Arguments**

- `individual`: Boolean parameter for feature-specific linear layers

All other parameters (e.g batch size, normalization scheme, etc...) are similar to the ones examined for Seq2Seq

## A.4  SCALING ANALYSIS OF THE PROPOSED MODEL

To further evaluate the scalability of our model, we include two additional plots showcasing its performance with respect to validation loss and inference time as the number of parameters increases.

- **Inference Time Scaling:** The first plot examines the inference time versus model size. The Seq2Seq+ASL model achieves orders-of-magnitude lower inference times compared to the Transformer baseline, highlighting its computational efficiency, crucial when intended to be used on an edge device
- **Validation Loss Scaling:** The second plot demonstrates how the validation mean absolute error (MAE) scales with model size. Notably, the Seq2Seq+ASL model consistently shows improved MAE performance as the number of parameters increases, generally outperforming the Transformer baseline on validation loss across all configurations.

## A.5  COMPARISON TO CURRENT STATE-OF-ART METHODS

We compare the following five models: **Seq2Seq** is a sequence-to-sequence model introduced by Bahdanau *et al.* (2014) Bahdanau et al. (2014), which is widely used for sequence prediction tasks. **Transformer**, introduced by Vaswani *et al.* (2017) Vaswani et al. (2023), is a powerful architecture based on self-attention mechanisms, showing promising results in various sequence modeling tasks. **AutoFormer** Wu et al. (2021) and **FedFormer** Zhou et al. (2022) are recent advancements in transformer models, designed on top of the transformer architecture suggesting auto-correlation layers, etc. However, the input and output dimensions of these models must match, which was not the case in our study (kinematic output size = 3, input forces = 4), requiring an additional linear layer to adjust the dimensions. This adjustment may have affected the performance of these models. **NLinear**, proposed by Zeng *et al.* (2023) Zeng et al. (2023), is a linear layer that incorporates normalization,

| Seq2Seq +ASL Parameter | Our Dataset | Open Source Dataset |
|---|---|---|
| `train_percent` | 0.75 | 0.75 |
| `val_percent` | 0.1 | 0.1 |
| `feature_win` | 512 | 256 |
| `target_win` | 1 | 1 |
| `intersect` | 1 | 1 |
| `model_class_name` | "Seq2Seq" | "Seq2Seq" |
| `optimizer_name` | "Adam" | "Adam" |
| `criterion_name` | "L1Loss" | "L1Loss" |
| `patience` | 10 | 10 |
| `patience_tolerance` | 0.005 | 0.005 |
| `n_epochs` | 30 | 30 |
| `seed` | 3407 | 3407 |
| `features_norm_method` | "zscore" | "zscore" |
| `targets_norm_method` | "identity" | "identity" |
| `features_global_normalizer` | true | true |
| `targets_global_normalizer` | true | true |
| `model_args_enc_embedding_size` | 10 | 30 |
| `model_args_enc_hidden_size` | 110 | 100 |
| `model_args_enc_num_layers` | 1 | 1 |
| `model_args_enc_bidirectional` | false | false |
| `model_args_dec_output_size` | 3 | 3 |
| `model_args_use_asl` | true | true |
| `model_args_concat_asl` | false | false |
| `model_args_complexify` | false | false |
| `model_args_gate` | true | true |
| `model_args_multidim_fft` | false | false |
| `model_args_dropout` | 0.1 | 0.1 |
| `model_args_freq_threshold` | 210 | 200 |
| `model_args_per_freq_layer` | true | false |
| `model_args_cross_spectrum_density` | false | false |
| `model_args_dec_hidden_size` | 110 | 100 |
| `model_args_dec_embedding_size` | 10 | 30 |
| `model_args_input_dim` | [512, 512, 4] | [512, 256, 5] |
| `batch_size` | 512 | 512 |

Table 4: **Hyperparameters of the best models.** The hyper-parameters used to train the Seq2Seq+ASL model on the validation set of each dataset

aiming to improve the stability and convergence of the model while utilizing nothing but a linear layer and non-linear activation.

## A.6 DATA ALIGNMENT

To align the force and angle measurements, we first smoothed the force signal and identified the onset of the force measurement by checking the force difference between consecutive timestamps. Second, we smoothed the 3D trajectory signal from the camera tracking system and identified the onset of the wing motion by checking if the position of each marker on the wing in 3D space exceeded a predefined threshold. Once the start times of the force and wing motion were identified, we aligned the two vectors in time. This procedure was also verified manually for each event.

## A.7 MOVIE 1

Movie 1 shows a representative event measure in our experimental system. Similarly to Figs. 1e,d, the movie shows the raw images taken from the two fast cameras, as well as representation of the 3D triangulated positions of the three markers on the wing.

