# OpenReview forum: "A deep inverse-mapping model for a flapping robotic wing"
_ICLR.cc/2025/Conference — ICLR 2025 Poster_

### Official Review · Reviewer_uvVA · 2024-10-22

**Soundness:** 2
**Presentation:** 3
**Contribution:** 2
**Rating:** 5
**Confidence:** 3

**Summary:**

This paper presented an adaptive spectrum layer enhanced seq2sep deep learning framework to learn the inverse-mapping model of a flapping robotic wing. The employed adaptive spectrum layer is found to have the advantage of learning the periodic features in the frequency domain. Overall, the presentation is clear, and a real experimental test is conducted. However, some important improvements are further needed, mainly including the theoretical contributions and more practical tests.

**Strengths:**

1. Adaptive spectrum layer enhanced seq2sep learning framework.
2. Real experimental tests.
3. Clear presentation.

**Weaknesses:**

1. It seems the main contribution of this paper is the introduction of the adaptive spectrum layer in seq2sep prediction missions. Empirical effects are provided in the results (although the improvement provided in Table 2 seems not very promising). The innovation is relatively weak. The theoretical principle and intuitive explanation behind the learning architecture should be further abstracted and discussed.

2. The implemented experiments are relatively simple. The real applied scenarios are usually more complicated, such as external disturbance, unknown dynamic model, and actuator uncertainty. The generalization problem of the developed learning architecture is not considered.  To improve contributions, it is highly recommended to further include the learned inverse model in the online control framework, not just offline demonstration on the dataset of a flapping wing. Moreover, a comparison with a traditional control method (e.g., MPC) is needed.

3. Some presentation errors. Every abbreviation that appears first in an article should be given its full name, such as FFT. All symbols employed in Fig 3 should be illustrated.

**Questions:**

1. The equation (2) is called the one-step-ahead prediction model. Does \tau refer to the one-step size? However, \tau in equation (1) is defined as a period. It is confused.

2. Through the learned inverse model, the desired attitude can be obtained from the desired force. In real application, how to plan the desired force and how to implement the obtained attitude?  A special control framework can be provided to illustrate the implementation of the learned inverse model. It is wondered if the uncertainty that exists in the control loop would influence the learning performance.

---

> ### Author Response · Authors · 2024-11-26
>
> We thanks the Reviewer.
>
> Comment: It seems the main contribution of this paper is the introduction of the adaptive spectrum layer in seq2sep prediction missions. Empirical effects are provided in the results (although the improvement provided in Table 2 seems not very promising). The innovation is relatively weak. The theoretical principle and intuitive explanation behind the learning architecture should be further abstracted and discussed.
>
> Answer: The innovation of this work is not only the introduction of the adaptive spectrum later, but also in the formation and learning of a new problem – an inverse mapping problem – in the context of an experimental system with complex and nonlinear dynamics. The improvement upon state-of-the-art architectures, such as Transformer, are sizeable (cf. our response to the review above), and obtained with a simpler architecture with improved scaling and latency (cf. response above).
>
> Per the Reviewer’s suggestion, we provide an intuitive explanation for the learning architecture in Section 3:
>
> “The essence of inverse-mapping modeling in this system lies in learning the relationship between desired output aerodynamic forces and the input wing kinematics that generate them.
> Intuitively, the input to the system is a time-sequence of desired aerodynamic forces that the wing should generate. The output is the full wing kinematics that, when applied to the wing, would result in the desired forces. The system learns the inverse mapping in two parallel backbones: learning the time-dependent (Seq2Seq) and frequency dependent (ASL) relationships between the forces. As a result, both the timestamps and frequencies that are relevant for the mapping are learned”.
>
> Comment: The implemented experiments are relatively simple. The real applied scenarios are usually more complicated, such as external disturbance, unknown dynamic model, and actuator uncertainty. The generalization problem of the developed learning architecture is not considered. To improve contributions, it is highly recommended to further include the learned inverse model in the online control framework, not just offline demonstration on the dataset of a flapping wing. Moreover, a comparison with a traditional control method (e.g., MPC) is needed.
>
> Answer: Unfortunately, the experiments performed here are not simple at all. Especially, while controlling the motor may appear trivial, it is utterly not so in these frequencies and accuracy constraints. While we would have loved to implement our system in a full-scale flying device, this task is complicated enough to place it beyond the scope of this work. Here, we demonstrate a critical and new module of a control system for flapping-wing systems that solves a NEW problem posed as an inverse mapping problem. Dealing with the many challenges suggested above would be the focus of future work.
>
> Comment: Some presentation errors. Every abbreviation that appears first in an article should be given its full name, such as FFT. All symbols employed in Fig 3 should be illustrated.
> A: Corrected. with thanks.
>
> Q: The equation (2) is called the one-step-ahead prediction model. Does \tau refer to the one-step size? However, \tau in equation (1) is defined as a period. It is confused.
> A: In both eqs. 1 and 2, \tau is the window size, as defined in the sentence following eq. 1. Specifically, \tau is size of the look-back window, which is the the model input, as usually defined by conditional probabilities (eqs. 1,2).
>
> Q: ...In real application, how to plan the desired force and how to implement the obtained attitude? A special control framework can be provided to illustrate the implementation of the learned inverse model. It is wondered if the uncertainty that exists in the control loop would influence the learning performance.
>
> A: In a flying flapping-robot, the desired force would be determined by the flight controller. In the currently existing robots, this stage is performed implicitly, via simple and manually-tuned PID control loops, whose output is wing kinematics, without directly determining the force (cf. Introduction). This implicit and heuristic approach does not exploit the full performance envelope of these systems, which pales in comparison to the performance of real insects. Namely, as we learn from observing insect flight, flapping wing systems have much broader capabilities and we believe that the current robots’ maneuverability, agility and robustness could be significantly improved using our explicit inverse-mapping. Our system will be combined as a module in such a flight controller (cf. Conclusion Section).  Providing “a special control framework” is currently beyond the scope of this work – it is very challenging and we haven’t yet developed a fully flying device to test it on.  As explained in the paper, the concept of inverse mapping is not limited to flapping-wing drones and translated to other systems as well.

---

> > ### Comment · Reviewer_uvVA · 2024-11-28
> > **Thanks for the reply**
> >
> > Thanks for the reply！ Some confused presentation problems have been addressed. However, the theoretical contribution and practical/mature experiments that are concerned haven't been improved. I will keep my score.

---

### Official Review · Reviewer_kJ1A · 2024-10-29

**Soundness:** 4
**Presentation:** 3
**Contribution:** 3
**Rating:** 8
**Confidence:** 4

**Summary:**

This paper presents a novel method for mapping the complex dynamics of a wing flapping to output thrust. The method employs a sequence to sequence model with a GRU encoder and decoder. The authors present a novel layer dubbed the Adaptive Sequence Layer (ASL) to attend to features across the frequency spectrum of the input sequence. The authors use two datasets: one is created by building a mechanical flapping wing and the second is an open source wing flapping dataset. The Seq2Seq+ASL method outperforms baseline methods such as Seq2Seq without ASL and a Transformer. The paper makes a clear and useful contribution to literature.

**Strengths:**

The paper has a number of strengths. The experimental setup is unique and interesting to the robotics community. The authors present a novel layer Adaptive Spectrum Layer (ASL), which in the experiments section is shown to improve the overall prediction performance. The choice of baselines are appropriate. The paper is well presented and clear to follow. The paper is significant.

**Weaknesses:**

Despite the strengths of the paper, there are a few weaknesses, but these should be easily addressed. Some of the key figures such as Fig. 1 and 2 are quite small. A recommendation would be to increase the size of these at the expense of some of the text of by resizing Fig. 5. For example, the abstract and introduction are a little on the verbose side. Nevertheless, these paragraphs are clear.

There are many comparisons between different baseline methods, which is good. However, it would be beneficial to have a statistical test to show that the improvement in performance between Seq2Seq+ASL and the other baseline methods is statistically significant. In Figure 5, there is not that much difference between the Seq2Seq and Transformer models. A statistical test such as Mann Whitney U-test to compare the differences between the MAE losses. I leave the choice of statistical test to the authors discretion.

**Questions:**

There are a few questions I have for the authors and these are included below.

Please would the authors perform the statistical test to compare the statistical differences between MAE of their method against the baselines.

Since ASL is a novel contribution of the authors, I would emphasise this to a greater degree in the abstract and the introduction. Is there a reason for not doing this?

Please could you point me to where you state the size of the dataset and evaluation sets used to compare results against the baselines.

I have a further optional suggestion for authors. Can the authors comment on the frequency spectrum present in the data?

This question is more one of curiosity, but may be of interest to other readers: I have a question about the RFFT and the IRFFT. How do you implement this in practice? I had assumed that most “fast” implementations are non-differentiable and I am curious which implementation you used.

---

> ### Author Response · Authors · 2024-11-26
>
> We thank the Reviewer for noting the novelty, uniqueness and significance of this work. Below we refer to each on the the Reviewer's comments and suggestions.
>
> Q: Figures.
> A: Figures were edited and are now more readable. Thank you.
>
>
> Comment: There are many comparisons between different baseline methods, which is good. However, it would be beneficial to have a statistical test to show that the improvement in performance between Seq2Seq+ASL and the other baseline methods is statistically significant. In Figure 5, there is not that much difference between the Seq2Seq and Transformer models. A statistical test such as Mann Whitney U-test to compare the differences between the MAE losses. I leave the choice of statistical test to the authors discretion.
>
> A: We thank the referee for this very insightful comment.
>
> To provide more statistical power, we used the Wilcoxon signed rank test. Unlike the Mann Whitney U-Test, this test does not assume that the samples are independently distributed, which is more suitable for our case, since predictions were made on the same dataset. According to the Wilcoxon test, on our dataset, our system outperformed the Transform with a p-value of 0.06. Still, we note that our system outperformed the Transformer in 2/3 of the cases in our dataset, which is an indication for its performance.  On the open-source dataset, our system outperformed the Transformer in 54.2% of the cases. The Wicoxon p-value in this case was 0.58, consistent with the notion these two architectures are equivalent on this dataset. These results have been added to the main text.
>
> Q: Please would the authors perform the statistical test to compare the statistical differences between MAE of their method against the baselines.
> A: Done, as detailed above.
>
> Q: Since ASL is a novel contribution of the authors, I would emphasize this to a greater degree in the abstract and the introduction. Is there a reason for not doing this?
> A: Corrected, with thanks.
>
> Q: Please could you point me to where you state the size of the dataset and evaluation sets used to compare results against the baselines.
> A: The sizes of the datasets are given in Table 1. The division is given in Line 418 of the manuscript: “Both datasets were randomly divided such that 75% of the events were used as a training set, 10% reserved for validation, and 15% for testing.”
>
> Q: I have a further optional suggestion for authors. Can the authors comment on the frequency spectrum present in the data?
> A: A sample spectrum is shown in the SI. Most power is concentrated at <210Hz.
>
> Q: This question is more one of curiosity, but may be of interest to other readers: I have a question about the RFFT and the IRFFT. How do you implement this in practice? I had assumed that most “fast” implementations are non-differentiable and I am curious which implementation you used.
> A: There are built-in differentiable layers in PyTorch for FFT/IFFT and RFFT/IRFFT, which we used here. The specific implementation will be available on GitHub upon publication of this work.

---

> > ### Comment · Reviewer_kJ1A · 2024-11-28
> >
> > Thank you for your responses and your updated manuscript. This reviewer is happy to increase their rating of the paper.
> >
> > P.s. Please add a space between control Ling et al. on line 150.

---

### Official Review · Reviewer_9m96 · 2024-11-03

**Soundness:** 3
**Presentation:** 3
**Contribution:** 3
**Rating:** 5
**Confidence:** 4

**Summary:**

This paper introduces a machine learning approach to control flapping-wing robots by developing a model that determines how wings should move to achieve desired forces. The key innovation is combining a sequence-to-sequence neural network with a new Adaptive Spectrum Layer (ASL) that better handles periodic motions. Tested on experimental data, the approach shows 11% improvement over existing methods and provides practical real-time control capabilities.

**Strengths:**

**Originality:**
- Novel inverse mapping approach for flapping wing control
- Creative integration of frequency domain processing (ASL) with sequence learning
- New experimental setup combining force and motion measurements
- Innovative application of deep learning to fluid dynamics control

**Quality:**
- Rigorous experimental validation:
  * Two different datasets (air and viscous fluid)
  * Comprehensive ablation studies of ASL components
  * Clear performance metrics and comparisons
- Thorough implementation details:
  * Full hyperparameter specifications
  * Clear architectural choices
  * Reproducible results

**Clarity:**
- Well-structured presentation
- Clear problem formulation and motivation
- Detailed technical explanations with supporting figures
- Comprehensive supplementary materials
- Open-source data and framework

**Significance:**
- Practical impact:
  * Real-time capable control system
  * Improved performance (11% over state-of-art)
  * Direct application to existing robotic systems
- Broader implications:
  * Framework applicable to other complex dynamic systems
  * Potential applications in biomedical devices
  * Open datasets for future research
- Technical contributions:
  * New insights into frequency domain processing
  * Improved understanding of flapping wing dynamics
  * Efficient model architecture design

**Weaknesses:**

- typo L418: (forces vs. force and torque)
- paper mentions different measurement types between datasets without explaining impact on model or justification
- What is the sim2real gap for the real wing-driven robot? How to narrow the sim2real gap to make the research more useful.
- Can you scale to multiple degrees of freedom? how to evaluate the scaling?
- Can you scale to different geometry and material? How to evaluate?
- What are the flight conditions?
- Any analysis of frequency selection? Why 100Hz/210Hz?
- More implementation details could be provided: synchronization for different sensors, delay?
- How is the sensor data aligned between cameras (10,000 fps) and force sensors (5,000 samples/sec)?
- What's the real-time performance on actual hardware? Processing delays?
- Any stability analysis or guarantees for the control system?
- How does the system handle disturbances or noise?
- Is there any theoretical justification for the model architecture choices?
- How generalizable is this approach across different Reynolds numbers?
- How about the performance of some basic NN structures, like MLP/LSTM/RNN?

**Questions:**

One of the contribution is the whole pipeline to collect the data. As for the ASL structure, I am not sure the necessity of the complex network structure. One core aspect of the paper is modeling aerodynamics, and similar work exists in the UAV field, such as NeuralFly. Therefore, this core contribution or novelty needs to be better clarified by the authors.

---

> ### Author Response · Authors · 2024-11-26
>
> We thank the Reviewer for acknowledging the innovative aspects, implications and contribution of this work.
>
> Q: paper mentions different measurement types between datasets without explaining impact on model or justification
> A: The main difference between these dataset is their different sensor configurations and markedly different flow regimes, which our model learned successfully (Sections 4.1 and 4.2).
>
> Q: What is the sim2real gap for the real wing-driven robot? How to narrow the sim2real gap to make the research more useful.
> A: The term “sim2real gap” might not best describe the current setup, as it lacks any simulation. Learning was performed directly from experimental data. Beyond that, we expect minor differences between the desktop-mounted system and its airborne version, stemming from second order aerodynamic effects.
>
> Q: Can you scale to multiple degrees of freedom? how to evaluate the scaling?
> A: Our model deals with all 3 DOFs of a rigid wing connected at a hinge, while we actuate the stroke angle. In fact, most robotic applications control only phi, and the elevation angle is constrained to be zero, and wing-pitch angle is determined passively.
>
> Q: Can you scale to different geometry and material? How to evaluate?
> A: Our method handled data from two very different experimental systems, with different kinematics, materials and fluids, demonstrating its ability to scale across different systems.
>
> Q: What are the flight conditions?
> A: Estimated future flight parameters: flapping frequency ~20Hz, max speed ~5m/s, max rotation rate of ~200deg/sec, endurance time 5-10 minutes. Mass 30-50gr.
>
> Q: Any analysis of frequency selection? Why 100Hz/210Hz?
> A: These frequencies were based on examining the frequency content of the raw data and, particularly, the noise. All relevant information was contained in the band <210Hz (x10 of typical flapping frequency).
>
> Q: ... synchronization for different sensors, delay, data alignment
> A: An explanation added to Appendix A.6
>
> Q: ...Processing delays?
> A: Delay is ~milliseconds, acceptable for wingbeat period of tens of ms, since control commands are calculated once per cycle.
>
> Q: Any stability analysis or guarantees for the control system?
> A: We expect that this module will improve the control performance of existing FW-MAVs, which currently rely on manually-tuned control loops, typically of the PID type.
>
> Q: How does the system handle disturbances or noise?
> A: The system provides a nominal mapping between desired force/torque and the wing motion that generates them. On its own, the system is not designed to handle flight disturbances -- this is done by the flight controller. Rather, our system will provide to the controller an “interface” between desired forces/torques and actuated wing motion.
>
> Q: theoretical justification for the model architecture choices
> A: First, The relevant data are temporal sequences of forces and angles, which are most suitable for seq2seq architectures. Second, RNN-based models have linear forward-pass time complexity, unlike transformer-based architectures, which use self-attention mechanisms with quadratic time complexity. This consideration is crucial for low-resource deployed devices, which cannot run parallel multi-core computation. Third, adding representation in frequency space is instrumental to modeling the relation between force inputs and wing kinematics, as both of them are periodic with similar frequencies.
>
> Q: generalizing to different Reynolds
> A: Tested Re span 1,000-50,000: the relevant flow regime for medium to large insects and, most importantly, flapping-wing MAVs (Sec 4.1).
>
> Q: performance of some basic NN
> A: We examined two linear models: LINEAR and NLINEAR, which underperformed compared with our system (Table2/Fig.5).
>
> Q: ... One core aspect of the paper is modeling aerodynamics, and similar work exists in the UAV field, such as NeuralFly. Therefore, this core contribution or novelty needs to be better clarified by the authors.
> A: First, flapping wing aerodynamics is orders of magnitude more complex than quadcopters. The forward problem of flapping wing has been extensively studied, but NO previous paper studied the inverse mapping (from desired forces to kinematics). Yet, this is the relevant problem for practical flight control of FW-UAVs. Second, recent papers that solve various problems in controlling quadcopters using ML/RL, e.g. under structural damage and external perturbations. NeuraFly is a neat example, in which an adaptive system learns the residual forces required to mitigate windy conditions. Our system is fundamentally different because it calculated the inverse mapping - from forces to wing kinematics. This problem is almost nonexistent in quadcopters because each rotor has only one DOF -  rotation speed - and mapping between speed to force is trivial. Flapping wings are different -their aerodynamics is much more complex and hence the mapping. This point is emphasized in the revised intro.

---

### Official Review · Reviewer_Qmqi · 2024-11-04

**Soundness:** 3
**Presentation:** 3
**Contribution:** 3
**Rating:** 6
**Confidence:** 4

**Summary:**

This paper releases a deep learning architecture to model inverse dynamics of a flapping wing which addresses the challenge of controlling such complicated and intricate systems. They also developed an experimental setup to collect data on the wing motion using high speed cameras. Their model uses a sequence-to-sequence framework enhanced with a frequency domain layer for adaptive learning, outperformed baseline models on author collected test data.

**Strengths:**

1. Extensive hyperparameter search to get optimal results.
2. Data collection is commendable.
3. Valid ablation and limitation sections have been provided.

**Weaknesses:**

1. The results present in Table 2 does not present more significant information than what is present in figure 5. Instead the ablation results can be moved to the main manuscript from the supplementary material
2. The Seq-2-Seq ASL model does not outperform the transformer on the open source dataset. But does perform better on the authors dataset. An explanation for this would greatly help the contribution of this paper.
3. The abstract should clarify that the 11% improvement is over the median since over mean the model performs worse than the baselines.

**Questions:**

1. Around 1000 experiments were performed for the hyperparameter search of the Seq-2-Seq ASL model. Was the same search conducted for other models in Table 2 ?
2. What is the effect of scaling this model on the MAE ? Considering the goal of this effort is to deploy on highly compute constrained platforms, it would be interesting to see if this model scales better than a transformer.

---

> ### Author Response · Authors · 2024-11-26
>
> Comment Re. Figure 5 and Table 2.
>
> Answer: We appreciate the suggestion. When we tried to accommodate the ablation study data,  there was not enough room in the main text, especially following the post-review corrections. The value of Figure 5 is in showing the broader picture of the models performance, and Table 2, in our opinion, is a convenient summary of performance, in the format typical to the field.
>
> Comment: The Seq-2-Seq ASL model does not outperform the transformer on the open source dataset. But does perform better on the authors dataset. An explanation for this would greatly help the contribution of this paper.
>
> Answer: on the openSrc dataset, performance of the two models is equivalent in the mean metric: 0.1130 for our model vs. 0.1123 for the Transformer model. In the median metric, our model outperformed the Transformer by 11%. In this particular setting, Fig. 5 shows that seq2seq+ASL has more outliers than the Transformer, which explains the difference between the mean and median metrics. Clarification has been added to the caption 5.
>
> On our dataset, the seq2seq+ASL outperformed the Transformer by 10.4% (mean) and 15% (median). Relatively, our system outperformed the Transformer in 2/3 of the test samples.
>
> Equality important to the performance per-se, we note that our model inference is significantly faster than the Transformer model (new figure in Appendix + comment below). Thus, even though the models are equivalent on one of the two datasets, our model inference time makes it practical for onboard computing, much more than the Transformer.
>
> A new paragraph has been added to the Results section.
>
> Comment Re. the abstract
> Answer: Corrected as suggested.
>
> Question: Around 1000 experiments were performed for the hyperparameter search of the Seq-2-Seq ASL model. Was the same search conducted for other models in Table 2 ?
>
> Answer: Yes. Details about this procedure have been added to Section A3 in the revised Appendix.
>
> Question: What is the effect of scaling this model on the MAE ? Considering the goal of this effort is to deploy on highly compute constrained platforms, it would be interesting to see if this model scales better than a transformer.
>
> Answer: We thank the Reviewer for this question. A new figure showing the scaling of both the validation-MAE and the inference time with the number of parameters is now included in the Appendix (Section A.4). These results explain the selected configurations of seq2seq+ASL (200k parameters) and the Transformer model (50k parameters) analyzed in this work. Interestingly, while seq2seq+ASL has x4 more parameters than the Transformer, seq2seq+ASL has  x10 shorter inference time, which makes it more practical for operational use given the flapping frequencies of current FW-MAVs. Thank you.

---

> > ### Comment · Reviewer_Qmqi · 2024-11-30
> >
> > Thanks for the thorough response and updates to the paper. I'll be increasing the contribution -> 3, but I shall be keeping the rating of 6.

---

### Author Response · Authors · 2024-11-26

We thank the Reviewers for their time and comments. We addressed all of the comments and truly believe that this procedure has improved the manuscript. In particular, we note the new scaling analysis of the model size and inference runtime, which shows that although our model has x4 more parameters than the best Transformer model, our model is x10 times faster, making it more practical for onboard applications in flying robots. In terms of accuracy, our model is better or equivalent to the Transformer, depending on dataset and metric. Hence, combining accuracy and runtime, the new analyses, motivated by the Reviewers, show that our model is superior for the problem at hand.

Relevant changes in the revised manuscript are marked in blue.

---

### Meta-Review · Area_Chair_et4X · 2024-12-23

**Metareview:**

The paper proposes a deep learning-based model for representing the inverse dynamics of complex, flapping-wing robots. Integral to the model is its integration of a new adaptive spectrum layer into a sequence-to-sequence neural network to improve the model's handling of periodic motion. Training and evaluation are performed using two datasets, one created by the authors that building a mechanical flapping wing and the second being an open-source wing flapping dataset. Experimental results demonstrate that the proposed Seq2Seq+ASL method outperforms baseline methods including a sequence-to-sequeqnce model without ASL and a Transformer.

The paper was reviewed by four referees. At least three of the reviewers emphasized the clarity and quality of the writing, which makes paper easy to follow. Some reviewers noted the extensiveness of the experimental evaluation, while at least three reviewers appreciated the experimental setup that combines force and motion measurements. Among the concerns raised by the reviewers was that the proposed method does not outperform the Transformer baseline on the open-source dataset (the difference between the two methods is not statistically significant), while it did outperform the baseline on the authors' custom dataset. That said, while the two methods perform similarly on the open-source dataset, the inference speed of the proposed method is significantly faster than that of the baseline. The paper would benefit from a more nuanced discussion that makes the contributions over the existing-state-of-the-art clearer.

**Additional Comments On Reviewer Discussion:**

The reviewers were concerned about the lack of a clear performance difference between the proposed method and the transformer baseline on the open-source dataset.

---

### Decision · Program_Chairs · 2025-01-22

Accept (Poster)